# Interpreting diffusion score matching using normalizing flow

**Wenbo Gong** [*1]   **Yingzhen Li** [*2]

## Abstract

Scoring matching (SM), and its related counterpart, Stein discrepancy (SD) have achieved great success in model training and evaluations. However, recent research shows their limitations when dealing with certain types of distributions. One possible fix is incorporating the original score matching (or Stein discrepancy) with a diffusion matrix, which is called *diffusion score matching* (DSM) (or *diffusion Stein discrepancy* (DSD)) . However, the lack of the interpretation of the diffusion limits its usage within simple distributions and manually chosen matrix. In this work, we plan to fill this gap by interpreting the diffusion matrix using normalizing flows. Specifically, we theoretically prove that DSM (or DSD) is equivalent to the original score matching (or score matching) evaluated in the transformed space defined by the normalizing flow, where the diffusion matrix is the inverse of the flow's Jacobian matrix. In addition, we also build its connection to Riemannian manifolds, and further extend it to continuous flows, where the change of DSM is characterized by an ODE.

## 1. Introduction

Recently, score matching (Hyvärinen, 2005) and its closely related counterpart, Stein discrepancy (Gorham, 2017) have made great progress in both understanding their theoretical properties and practical usage. Particularly, unlike Kullback–Leibler (KL) divergence which can only be used for distributions with known normalizing constant, SM (or SD) can be evaluated for unnormalized densities, and requires fewer assumptions for the probability distributions (Fisher et al., 2021). Such useful properties enable them to be widely applied in training energy-based model (EBM) (Song

et al., 2020a; Grathwohl et al., 2020; Wenliang et al., 2019), state-of-the-art score-based generative model (Song & Ermon, 2019; Song et al., 2020b), statistical tests (Liu et al., 2016; Chwialkowski et al., 2016) and variational inference (Hu et al., 2018; Liu & Wang, 2016).

Despite their elegant statistical properties, recent work (Barp et al., 2019) demonstrated their failure when dealing with certain type of distributions (e.g. heavy-tailed distributions). For instance, when the data and the model are heavy tailed distributions, the model can fail to recover the true mode even in one dimensional case. The root of this problem is that the SM (or SD) objective is highly non-convex and does not correlate well with likelihood. To fix it, Barp et al. (2019) proposed a variant called *diffusion score matching* (and *diffusion Stein discrepancy*), where a diffusion matrix is introduced. However, the author did not provide us an interpretation of this diffusion matrix. In fact, the diffusion used by the author (Barp et al., 2019) is manually chosen for toy densities. Such lack of interpretation hinders further development of a proper training method of the diffusion matrix.

In this paper, we aim to give an interpretation based on normalizing flows, which sheds light on developing training method for the diffusion. We summarize our contributions as follows:

- We theoretically prove that DSM (or DSD) is equivalent to the original SM (or SD) performed in the transformed space defined by the normalizing flow. The diffusion matrix is exactly the same as the inverse of the flow's Jacobian matrix.

- We further show that its connection to Riemannian manifold. Specifically, we show the diffusion matrix is closely related to the Riemannian metric tensor.

- We further extend DSM to their continuous version. Namely, we derive an ODE to characterize its instantaneous change.

We hope that by building these connections, a broad range of techniques from normalizing flow communities can be leveraged to develop training methods for the diffusion matrix.

---

[*]Equal contribution  [1]Department of Engineering, University of Cambridge, Cambridge, UK [2]Department of Computing, Imperial College London, London, UK. Correspondence to: Wenbo Gong <wg242@cam.ac.uk>.

Third workshop on *Invertible Neural Networks, Normalizing Flows, and Explicit Likelihood Models* (ICML 2021). Copyright 2021 by the author(s).

## 2. Background: Diffusion Stein discrepancy

### 2.1. Score matching and Stein discrepancy

Let $\mathcal{P}$ be the space of Borel probability measures on $\mathbb{R}^D$, $\mathbb{Q} \in \mathcal{P}$ to be a probability measure, the objective for model learning is to find a sequence of probability measures $\{\mathbb{P}_\theta : \theta \in \Theta\} \subset \mathcal{P}$ that approximates $\mathbb{Q}$ in an appropriate sense. One common way to achieve this is by defining a discrepancy measure $\mathcal{D} : \mathcal{P} \times \mathcal{P} \to \mathbb{R}$, which quantifies the differences between two probability measures. Thus, the optimal parameters $\theta^*$ can be obtained by $\theta^* = \mathrm{argmin}\mathcal{D}(\mathbb{Q}||\mathbb{P}_\theta)$. The choice of discrepancy depends on the properties of the probability measures, the efficiency and its robustness. The one we are focused on is called *Fisher divergence*. Assuming for probability measures $\mathbb{Q}$ and $\mathbb{P}_\theta$, we have corresponding twice differentiable densities $q(\boldsymbol{x})$, $p_\theta(\boldsymbol{x})$. The *Fisher divergence* (Johnson, 2004) is defined as

$$\mathcal{F}(q, p) = \frac{1}{2}\mathbb{E}_q[||\boldsymbol{s}_p(\boldsymbol{x}) - \boldsymbol{s}_q(\boldsymbol{x})||^2] \quad (1)$$

where $\boldsymbol{s}_p(\boldsymbol{x}) = \nabla_{\boldsymbol{x}} \log p_\theta(\boldsymbol{x})$ is called the score of $p_\theta$, and $\boldsymbol{s}_q$ is defined accordingly. Despite that $q$ is often used for underlying data densities with the intractable $\boldsymbol{s}_q$, $\boldsymbol{s}_q$ in fact acts as a constant for parameter $\theta$. Thus, one can use integration-by-part to derive the following:

$$\mathcal{F}(q, p_\theta) = \underbrace{\mathbb{E}_q\left[\frac{1}{2}||\boldsymbol{s}_p(\boldsymbol{x})||^2 + Tr(\nabla_{\boldsymbol{x}}\boldsymbol{s}_p(\boldsymbol{x}))\right]}_{SM(q, p_\theta)} + C_q \quad (2)$$

with $C_q$ a constant w.r.t. $\theta$. This equivalent objective $SM(q, p_\theta)$ is referred as *score matching* (Hyvärinen, 2005).

Another discrepancy measure we are interested is called *Stein discrepancy*, which is defined as

$$\mathcal{S}(q, p_\theta) = \sup_{\boldsymbol{f} \in \mathcal{H}} \mathbb{E}_q[\boldsymbol{s}_p(\boldsymbol{x})^T \boldsymbol{f}(\boldsymbol{x}) + \nabla_{\boldsymbol{x}}^T \boldsymbol{f}(\boldsymbol{x})] \quad (3)$$

where $\boldsymbol{f} : \mathbb{R}^D \to \mathbb{R}^D$ is a test function, and $\mathcal{H}$ is an appropriate test function family, e.g. reproducing kernel Hilbert space (Liu et al., 2016; Chwialkowski et al., 2016) or Stein class (Gorham, 2017; Liu et al., 2016). Recent work (Hu et al., 2018) proved a connection between Stein discrepancy and Fisher divergence by showing the optimal test function:

$$\boldsymbol{f}^*(\boldsymbol{x}) \propto \boldsymbol{s}_p(\boldsymbol{x}) - \boldsymbol{s}_q(\boldsymbol{x}). \quad (4)$$

thus we can show Stein discrepancy is equivalent to Fisher divergence up to a multiplicative constant.

Barp et al. (2019); Gorham et al. (2019) further extend the score matching and Stein discrepancy by incorporating a diffusion matrix $\boldsymbol{m}(\boldsymbol{x}) : \mathbb{R}^D \to \mathbb{R}^{D \times D}$. It starts from defining *diffusion Fisher divergence*

$$\mathcal{F}_m(q, p_\theta) = \frac{1}{2}\mathbb{E}_q[||\boldsymbol{m}(\boldsymbol{x})^T(\boldsymbol{s}_p(\boldsymbol{x}) - \boldsymbol{s}_q(\boldsymbol{x}))||^2] \quad (5)$$

where $\boldsymbol{m}(\boldsymbol{x})$ is a matrix-valued function. Expanding eq. 5 and applying integration by parts (with $\boldsymbol{m}$ short-handing $\boldsymbol{m}(\boldsymbol{x})$ and $\boldsymbol{s}_p$ short-handing $\boldsymbol{s}_p(\boldsymbol{x})$):

$$\mathcal{F}_m(q, p_\theta) = \underbrace{\mathbb{E}_q\left[\frac{1}{2}||\boldsymbol{m}^T\boldsymbol{s}_p||^2 + \nabla^\top(\boldsymbol{m}\boldsymbol{m}^\top\boldsymbol{s}_p)\right]}_{DSM_m(q, p_\theta)} + C_{q,m},$$
$$(6)$$

where $C_{q,m}$ depends on both $q$ and $\boldsymbol{m}(\boldsymbol{x})$. Similar to the derivation of $SM(q, p_\theta)$, this also returns an alternative diffusion score matching (DSM) objective $DSM_m(q, p_\theta)$.

Similarly, Diffusion Stein discrepancy (DSD) is defined as

$$DSD_m(q, p_\theta)$$
$$= \sup_{\boldsymbol{f} \in \mathcal{H}} \mathbb{E}_q[(\boldsymbol{m}(\boldsymbol{x})^T\boldsymbol{s}_p(\boldsymbol{x}))^T\boldsymbol{f}(\boldsymbol{x}) + \nabla_{\boldsymbol{x}}^T(\boldsymbol{m}(\boldsymbol{x})\boldsymbol{f}(\boldsymbol{x}))] \quad (7)$$

It can be shown that as long as $\boldsymbol{m}(\boldsymbol{x})$ is invertible, $\mathcal{F}_m(q, p_\theta)$ and $DSD_m(q, p_\theta)$ are valid divergences. These two extensions have demonstrated superior performances when dealing with certain type of distributions. In the following, we give a motivating example similar to Barp et al. (2019).

### 2.2. Motivating example: Student-t distribution

Let assume $q, p_\theta$ to be 1 dimensional student-t distribution. The target is to approximate $q$ by $p_\theta$. The training set is 300 i.i.d data sampled from $q$ with mean 0 and scale 0.3. We assume the scale parameter for $p_\theta$ is the same as $q$, and the only trainable parameter $\theta$ is the mean. The degree of freedom is 5 for both $q, p_\theta$.

The left panel of figure 1 shows the score matching loss computed for different $\theta$. We can observe that for original $SM(q, p_\theta)$ loss, it is highly non-convex, and the loss value does not correlate well with likelihood. Indeed, we can see the true location $\theta = 0$ is protected by two high 'walls'. In other words, unlike maximum likelihood estimator, a parameter $\theta$ that is closer to the ground truth does not necessarily produce low SM loss. One important consequence is that unless the initialized $\theta$ is within the narrow valid region, the gradient-based optimization will never recover the truth.

On the other hand, the middle panel of figure 1 shows that if we chose $\boldsymbol{m}(\boldsymbol{x}) = (1 + \frac{(\boldsymbol{x}-\theta)^2}{0.6})$ (Manual Flow) as the diffusion matrix, the corresponding $DSM_m(q, p_\theta)$ loss is convex. The ground truth can be recovered by minimizing DSM with a proper gradient-based optimizer.

However, this $DSM_m(q, p_\theta)$ is only a surrogate objective for learning $\theta$ because the diffusion matrix $\boldsymbol{m}$ contains $\theta$. Thus, the dropped term $C_{q,m}$ in eq.6 is no longer a constant. Although one can treat the $\theta$ in $\boldsymbol{m}$ as constant during training and ignore its contribution when taking the derivative, this is equivalent to use different losses after each $\theta$ update. We leave its convergence analysis for the future work.

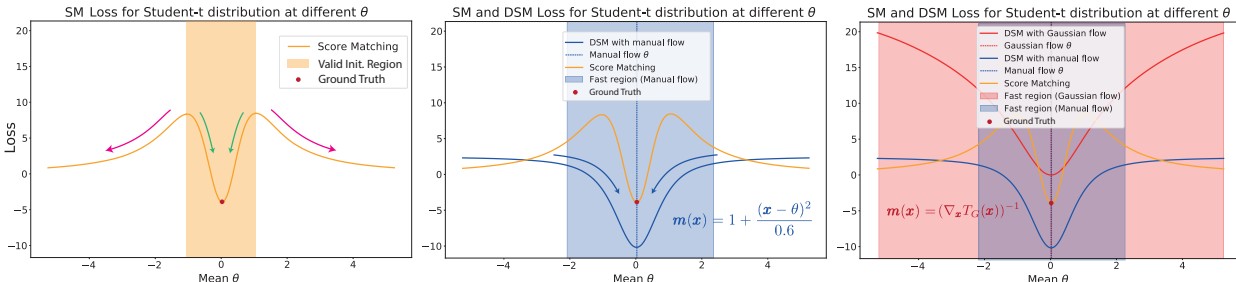

*Figure 1.* The $SM(q, p_\theta)$ and $DSM_m(q, p_\theta)$ losses computed with different mean parameters $\theta$. **Left**: This orange line plots the vanilla SM loss between $q$ and $p_\theta$. The arrow indicates the gradient descent direction of $\theta$. The red dot ● is the ground truth for $\theta$. **Middle**: The blue line plots the DSM loss with $\boldsymbol{m}(\boldsymbol{x}) = 1 + \frac{(\boldsymbol{x}-\theta)^2}{0.6}$. The blue rectangle indicates the region with large gradient descent magnitude (fast convergence). **Right**: The red line plots the DSM loss with Gaussian flow. The red rectangle indicates the fast convergence region.

The selection of the diffusion matrix is crucial to the success of the estimator. Unfortunately, the interpretation of this matrix is unclear, not mentioning a selection algorithm. In the following, we aim to shed lights on this problem by connecting this diffusion with normalizing flows.

## 3. Diffusion matrix as normalizing flow

### 3.1. Interpreting DSM/DSD using normalizing flow

Let assume we have two densities $q_X(\boldsymbol{x})$, $p_X(\boldsymbol{x})$ defined on $\mathbb{R}^D$ and are twice differentiable. We further define an differentiable invertible transformation $\boldsymbol{T}(\boldsymbol{x}) : \mathbb{R}^D \to \mathbb{R}^D$:

$$\boldsymbol{y} = \boldsymbol{T}(\boldsymbol{x}) \tag{8}$$

with the corresponding induced densities $q_Y(\boldsymbol{y})$ and $p_Y(\boldsymbol{y})$. We can prove the following theorem:

**Theorem 3.1.** *For twice differentiable densities $q_X(\boldsymbol{x})$, $p_X(\boldsymbol{x})$ and an invertible differentiable transformation $T : \mathbb{R}^D \to \mathbb{R}^D$, the diffusion Fisher divergence (Eq.5) is equivalent to the original Fisher divergence in $\boldsymbol{y}$ space:*

$$\mathcal{F}(q_Y, p_Y) = \frac{1}{2}\mathbb{E}_{q_Y}[||\boldsymbol{s}_{p_Y}(\boldsymbol{y}) - \boldsymbol{s}_{q_Y}(\boldsymbol{y})||^2] \tag{9}$$

*where $\boldsymbol{y} = T(\boldsymbol{x})$, and $p_Y$, $q_Y$ are corresponding densities after the transformation. The diffusion matrix $\boldsymbol{m}(\boldsymbol{x})$ is the inverse of the Jacobian matrix $(\nabla_{\boldsymbol{x}}\boldsymbol{T}(\boldsymbol{x}))^{-1}$*

*Proof.* From the change of variable formula, the corresponding densities $p_Y(\mathbf{y})$, $q_Y(\mathbf{y})$ can be defined as:

$$p_Y(\mathbf{y}) = p_X(T^{-1}(\mathbf{y}))|\frac{\partial T^{-1}(\mathbf{y})}{\partial \mathbf{y}}|,$$

$$q_Y(\mathbf{y}) = q_X(T^{-1}(\mathbf{y}))|\frac{\partial T^{-1}(\mathbf{y})}{\partial \mathbf{y}}|.$$

Then the Fisher divergence $\mathcal{F}(q_Y, p_Y)$ is formulated as:

$$\mathcal{F}(q_Y, p_Y) := \frac{1}{2}\mathbb{E}_{q_Y}[||\nabla_{\mathbf{y}} \log p_Y(\mathbf{y}) - \nabla_{\mathbf{y}} \log q_Y(\mathbf{y})||_2^2]$$

$$= \frac{1}{2}\mathbb{E}_{q_Y}[||\nabla_{\mathbf{y}} \log p_X(T^{-1}(\mathbf{y})) - \nabla_{\mathbf{y}} \log q_X(T^{-1}(\mathbf{y}))||_2^2]$$

$$= \frac{1}{2}\mathbb{E}_{q_Y}[||\nabla_{\mathbf{y}}T^{-1}(\mathbf{y})^{\top}$$
$$(\nabla_{T^{-1}(\mathbf{y})} \log p_X(T^{-1}(\mathbf{y})) - \nabla_{T^{-1}(\mathbf{y})} \log q_X(T^{-1}(\mathbf{y})))||_2^2]$$

$$= \frac{1}{2}\mathbb{E}_{q_X}[||(\nabla_{\boldsymbol{x}}T(\boldsymbol{x}))^{-\top}(\nabla_{\boldsymbol{x}} \log p_X(\boldsymbol{x}) - \nabla_{\boldsymbol{x}} \log q_X(\boldsymbol{x}))||_2^2], \tag{10}$$

where the last step comes from changing the variable to $\boldsymbol{x} = T^{-1}(\mathbf{y})$ and noticing that $\nabla_{\mathbf{y}}T^{-1}(\mathbf{y}) = (\nabla_{\boldsymbol{x}}T(\boldsymbol{x}))^{-1}$ from the inverse function theorem. This objective coincides with the *diffusion Fisher divergence* (Eq.5). Importantly, $\mathcal{F}_m(q_X, p_X)$ is a valid divergence (i.e. $\mathcal{F}_m(p_X, q_X) = 0$ iff. $p_X = q_X$) when $m(\boldsymbol{x})$ is an invertible matrix for every $\boldsymbol{x}$. As normalising flow transformations naturally give invertible Jacobian matrices, we can easily extablish the connection $\mathcal{F}(q_Y, p_Y) = \mathcal{F}_m(q_X, p_X)$ with $m(\boldsymbol{x}) = (\nabla_{\boldsymbol{x}}T(\boldsymbol{x}))^{-1}$. $\square$

We also include the likelihood plots afte the transformation in Appendix D.

Similarly, we can prove the connections between DSD (Eq.7) and normalizing flow. The proof is in appendix A.

**Theorem 3.2.** *For twice differentiable densities $q_X(\boldsymbol{x})$, $p_X(\boldsymbol{x})$, an invertible differentiable transformation $\boldsymbol{T}(\boldsymbol{x}) : \mathbb{R}^D \to \mathbb{R}^D$ and differentiable test function in suitable test function family $\mathcal{H}$: $\boldsymbol{f} : \mathbb{R}^D \to \mathbb{R}^D \in \mathcal{H}$, the diffusion Stein discrepancy (Eq.7) is equivalent to the original Stein discrepancy*

$$\mathcal{S}(q_Y, p_Y) = \sup_{\boldsymbol{g} \in \mathcal{H}'} \mathbb{E}_{q_Y}[\boldsymbol{s}_{p_Y}(\boldsymbol{y})^T\boldsymbol{g}(\boldsymbol{y}) + \nabla_{\boldsymbol{y}}^T\boldsymbol{g}(\boldsymbol{y})] \tag{11}$$

*where $\boldsymbol{g}(\mathbf{y}) = \boldsymbol{f}(\boldsymbol{T}^{-1}(\mathbf{y}))$, $\mathcal{H}'$ is the corresponding function space for $\boldsymbol{g}$, $p_Y$ and $q_Y$ are transformed densities by $\boldsymbol{T}(\cdot)$. The diffusion matrix $\boldsymbol{m}(\boldsymbol{x}) = (\nabla_{\boldsymbol{x}}\boldsymbol{T}(\boldsymbol{x}))^{-1}$.*

Based on the above two theorems, we formally establish the connections between the *diffusion Fisher divergence/DSD* with normalizing flows. This gives us an interpretation of the diffusion matrix as the inverse of the Jacobian matrix defined by the flow.

## 3.2. Better flow design

Based on the interpretation, we try to give a better design for the diffusion matrix $\boldsymbol{m}$. Here, we design a flow $\boldsymbol{T}_G(\cdot)$ that transforms the Student-t distribution $p_\theta$ to a standard Gaussian $\mathcal{N}(0,1)$, which we named as **Gaussian flow**:

$$\boldsymbol{T}_G(\boldsymbol{x}) = F_G^{-1} \circ F_\theta(\boldsymbol{x}). \tag{12}$$

Here $F_G$ and $F_\theta$ are cumulative density functions for $\mathcal{N}(0,1)$ and $p_\theta$ respectively. We plot the corresponding $DSM_m(q, p_\theta)$ loss in the right panel of Figure 1. Both the manually designed flow and Gaussian flow can recover the ground truth $\theta$ regardless of initialization. However, Gaussian flow allows faster convergence during training. The fast convergence regions is the region where the gradient of the DSM w.r.t $\theta$ has a magnitude greater than 1. The Gaussian flow has a much wider region compared to manual flow. The length of the region is 4.44 and 10.56 respectively (more than 2 times). For high dimensional distributions, this area of the region can scale up with $O(2^D)$, which can have significant impact on convergence speed. Another advantage of this systematic design of the diffusion matrix is its robustness, which is further discussed in Appendix E.

## 3.3. Interpreting DSM using Riemannian manifold

Assume we have a Riemannian manifold $(\mathcal{M}, \boldsymbol{g})$ with Riemannian metric tensor $\boldsymbol{g}$. For each point $\boldsymbol{a} \in \mathcal{M}$, we assume it has a local coordinates $\boldsymbol{x}_a = [x_a^1, \ldots, x_a^D]$. We can prove the following proposition:

**Proposition 3.1.** *Define two probability measures $\mathbb{Q}, \mathbb{P}$ on the Riemannian manifold $(\mathcal{M}, \boldsymbol{g})$ as defined above. We denote the corresponding densities (in terms of local coordinates $\boldsymbol{x}$) w.r.t. Riemannian manifold as $\tilde{p}(\boldsymbol{x}) = \frac{d\mathbb{P}}{d\mathcal{M}(\boldsymbol{x})}$ and $\tilde{q}(\boldsymbol{x}) = \frac{d\mathbb{Q}}{d\mathcal{M}(\boldsymbol{x})}$. Then, the Fisher divergence from $\tilde{q}$ to $\tilde{q}$ is*

$$\mathcal{F}_{\mathcal{M}}(\tilde{q}, \tilde{p}) = \frac{1}{2}\mathbb{E}_q[\boldsymbol{\Delta}(\boldsymbol{x})^T \boldsymbol{G}(\boldsymbol{x})^{-1} \boldsymbol{\Delta}(\boldsymbol{x})] \tag{13}$$

*where $p(\boldsymbol{x}) = \frac{d\mathbb{P}}{d\mathcal{M}(\boldsymbol{x})}\frac{d\mathcal{M}(\boldsymbol{x})}{d\boldsymbol{x}}$, $q(\boldsymbol{x})$ is defined similarly, and $\boldsymbol{\Delta}(\boldsymbol{x}) = \boldsymbol{s}_p(\boldsymbol{x}) - \boldsymbol{s}_q(\boldsymbol{x})$. $\boldsymbol{G}(\boldsymbol{x})$ is an symmetric positive definite matrix representing the Riemannian metric tensor. Particularly, if $\boldsymbol{G}(\boldsymbol{x}) = \boldsymbol{m}(\boldsymbol{x})^{-T}\boldsymbol{m}(\boldsymbol{x})^{-1}$, then $\mathcal{F}_{\mathcal{M}}(\tilde{q}, \tilde{p})$ is equivalent to the diffusion Fisher divergence (Eq.5) with diffusion matrix $\boldsymbol{m}(\boldsymbol{x})$.*

The proof is in appendix B.

This result is more general than theorem 3.1. Specifically, theorem 3.1 only proves a sufficient condition for the *diffu-*

*sion Fisher divergence* to be a valid discrepancy. Namely, if we have an invertible flow, the diffusion matrix $\boldsymbol{m}(x)$ must be invertible. However, the converse is not true. On the other hand, proposition 3.1 only requires $\boldsymbol{m}(\boldsymbol{x})$ to be invertible, which is more general. Indeed, from the topological point of view, if we have an invertible and differentiable flow $\boldsymbol{T}$, then the transformed space (Riemannian manifold) is actually diffeomorphic to the original space (e.g. $\mathbb{R}^D$). Thus, this flow can be viewed as a special case of Gemici et al. (2016). But in general, Riemannian manifold may not be diffeomorphic to $\mathbb{R}^D$, which explains why theorem 3.1 is only a sufficient condition.

## 3.4. Continuous DSM with ODE flow

Previous sections assume a deterministic transformation $\boldsymbol{T}(\boldsymbol{x})$. Recent work has shown promising results for continuous flows characterised by an ODE (Chen et al., 2018; Grathwohl et al., 2018).

$$d\boldsymbol{x} = \boldsymbol{g}(\boldsymbol{x}(t))dt \tag{14}$$

where $\boldsymbol{g}(\boldsymbol{x}(t))$ is a deterministic drift that is uniformly Lipschitz continuous w.r.t. $\boldsymbol{x}$. We define $p_t$ and $q_t$ to be the corresponding densities for $\boldsymbol{x}(t)$. Inspired by Chen et al. (2018), we can characterise the instantaneous change of the score matching loss $\frac{d\mathcal{F}(q_t, p_t)}{dt}$ by the following proposition:

**Proposition 3.2.** *Let $p_t(\boldsymbol{x}(t))$, $q_t(\boldsymbol{x}(t))$ be two probability density functions, where $\boldsymbol{x}(t)$ is characterized by an ODE defined in eq.14. Assume $\boldsymbol{g}(\boldsymbol{x}(t))$ is uniformly Lipschitz continuous w.r.t. $\boldsymbol{x}(t)$. Then, the instantaneous change of score matching loss follows:*

$$\frac{d\mathcal{F}(q_t, p_t)}{dt} = -\frac{1}{2}\mathbb{E}_{q_t}[\boldsymbol{\Delta}(\boldsymbol{x})^T(\nabla_{\boldsymbol{x}}\boldsymbol{g}(\boldsymbol{x}) + \nabla_{\boldsymbol{x}}\boldsymbol{g}(\boldsymbol{x})^T)\boldsymbol{\Delta}(\boldsymbol{x})] \tag{15}$$

*where $\boldsymbol{\Delta}(\boldsymbol{x}) = \boldsymbol{s}_{p_t}(\boldsymbol{x}) - \boldsymbol{s}_{q_t}(\boldsymbol{x})$.*

The proof is in appendix C.

# 4. Conclusion

In this paper, we discuss the connections of the *diffusion score matching* and *diffusion Stein discrepancy* to normalizing flows. Specifically, we theoretically prove that the *diffusion Fisher divergence* (or DSD) is equivalent to performing the original *Fisher divergence* (or *Stein discrepancy*) on the transformed densities. The diffusion matrix $\boldsymbol{m}(\boldsymbol{x})$ is defined by the inverse of the flow's Jacobian matrix. We also establish the connection of *diffusion Fisher divergence* with densities defined on Riemannian manifolds. In the end, we extend the *diffusion Fisher divergence* by continuous flow, and derive an ODE characterizing its instantaneous changes. By building the connections, we hope to shed lights on developing training method for the diffusion matrix to enable the practical usage for large models.

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

## A. Proof of theorem 3.2

*Proof.* Let's first define the Stein operator as

$$\mathcal{S}_{p_Y}[\boldsymbol{g}] = \boldsymbol{s}_{p_Y}(\boldsymbol{y})^T \boldsymbol{g}(\boldsymbol{x}) + \nabla_{\boldsymbol{y}}^T \boldsymbol{g}(\boldsymbol{y}) \tag{16}$$

for the test function $\boldsymbol{g}(\boldsymbol{y})$ and density $p_Y(\mathbf{y})$. Thus, the Stein discrepancy can be rewritten as

$$\mathcal{S}(q_Y, p_Y) = \sup_{\boldsymbol{g} \in \mathcal{H}} \mathbb{E}_{q_Y}[\mathcal{S}_{p_Y}[\boldsymbol{g}]] \tag{17}$$

In the following, we will focus on the Stein operator. From the change of variable formula $\boldsymbol{y} = \boldsymbol{T}(\boldsymbol{x})$, we have

$$p_Y(\mathbf{y}) = p_X(\boldsymbol{T}^{-1}(\mathbf{y})) \left| \frac{\partial \boldsymbol{T}^{-1}(\mathbf{y})}{\partial \mathbf{y}} \right|, \quad \boldsymbol{g}(\mathbf{y}) = \boldsymbol{f}(\boldsymbol{T}^{-1}(\mathbf{y})) \tag{18}$$

Now we can rewrite the Stein operator:

$$\mathcal{S}_{p_Y}[\boldsymbol{g}] = \nabla_{\mathbf{y}} \log p_Y(\mathbf{y})^T \boldsymbol{g}(\mathbf{y}) + \nabla_{\mathbf{y}}^T \boldsymbol{g}(\mathbf{y})$$
$$= \nabla_{\mathbf{y}} \log p_X(\boldsymbol{T}^{-1}(\mathbf{y}))^T \boldsymbol{g}(\mathbf{y}) + (\nabla_{\mathbf{y}} \log \left| \frac{\partial \boldsymbol{T}^{-1}(\mathbf{y})}{\partial \mathbf{y}} \right|)^T \boldsymbol{g}(\mathbf{y})$$
$$+ \nabla_{\mathbf{y}} \boldsymbol{g}(\mathbf{y})$$
$$= \left[ (\nabla_{\mathbf{y}} \boldsymbol{T}^{-1}(\mathbf{y}))^T (\nabla_{\boldsymbol{T}^{-1}(\mathbf{y})} \log p_X(\boldsymbol{T}^{-1}(\mathbf{y}))) \right]^T \boldsymbol{g}(\mathbf{y})$$
$$+ \underbrace{(\nabla_{\mathbf{y}} \log \left| \frac{\partial \boldsymbol{T}^{-1}(\mathbf{y})}{\partial \mathbf{y}} \right|)^T \boldsymbol{g}(\mathbf{y})}_{\textcircled{1}}$$
$$+ Tr[(\nabla_{\mathbf{y}} \boldsymbol{T}^{-1}(\mathbf{y})) \nabla_{\boldsymbol{T}^{-1}(\mathbf{y})} \boldsymbol{f}(\boldsymbol{T}^{-1}(\mathbf{y}))] \tag{19}$$

The second equality is from the chain rule and definition of divergence operator $\nabla^T$. For the layout of the matrix calculus, we follow the column vector layout as the following: for a function $\boldsymbol{h} : \mathbb{R}^D \to \mathbb{R}$, and $\boldsymbol{f} : \mathbb{R}^D \to \mathbb{R}^N$, we have

$$\frac{\partial h(\boldsymbol{x})}{\partial \boldsymbol{x}} = \begin{bmatrix} \frac{\partial h(\boldsymbol{x})}{\partial x_1} \\ \vdots \\ \frac{\partial h(\boldsymbol{x})}{\partial x_D} \end{bmatrix}$$
$$\frac{\partial \boldsymbol{f}(\boldsymbol{x})}{\partial \boldsymbol{x}} = \begin{bmatrix} \frac{\partial f_1(\boldsymbol{x})}{\partial x_1} & \cdots & \frac{\partial f_1(\boldsymbol{x})}{\partial x_D} \\ \vdots & \vdots & \vdots \\ \frac{\partial f_N(\boldsymbol{x})}{\partial x_1} & \cdots & \frac{\partial f_N(\boldsymbol{x})}{\partial x_D} \end{bmatrix} \tag{20}$$

Now, we focus on $\textcircled{1}$ term:

$$\nabla_{\mathbf{y}} \log \left| \frac{\partial \boldsymbol{T}^{-1}(\mathbf{y})}{\partial \mathbf{y}} \right| = Tr[(\nabla_{\mathbf{y}} \boldsymbol{T}^{-1}(\mathbf{y}))^{-1} \nabla_{\mathbf{y}} \nabla_{\mathbf{y}} \boldsymbol{T}^{-1}(\mathbf{y})]$$
$$= Tr[\nabla_{\boldsymbol{x}} \boldsymbol{T}(\boldsymbol{x}) \nabla_{\mathbf{y}} (\nabla_{\boldsymbol{x}} \boldsymbol{T}(\boldsymbol{x}))^{-1}]$$
$$= Tr[\nabla_{\boldsymbol{x}} \boldsymbol{T}(\boldsymbol{x}) \nabla_{\mathbf{y}} \boldsymbol{T}^{-1}(\mathbf{y}) \nabla_{\boldsymbol{x}} (\nabla_{\boldsymbol{x}} \boldsymbol{T}(\boldsymbol{x}))^{-1}]$$
$$= Tr[\nabla_{\boldsymbol{x}} (\nabla_{\boldsymbol{x}} \boldsymbol{T}(\boldsymbol{x}))^{-1}] \tag{21}$$

where we use the inverse function theorem $\nabla_{\mathbf{y}} \boldsymbol{T}^{-1}(\mathbf{y}) = (\nabla_{\boldsymbol{x}} \boldsymbol{T}(\boldsymbol{x}))^{-1}$. In addition, we define $\nabla_{\boldsymbol{x}}^T (\nabla_{\boldsymbol{x}} \boldsymbol{T}(\boldsymbol{x}))^{-1} = Tr[\nabla_{\boldsymbol{x}} (\nabla_{\boldsymbol{x}} \boldsymbol{T}(\boldsymbol{x}))^{-1}]$.

So we can set $\boldsymbol{m}(\boldsymbol{x}) = (\nabla_{\boldsymbol{x}} \boldsymbol{T}(\boldsymbol{x}))^{-1}$, we can obtain:

$$\mathcal{S}_{p_Y}[\boldsymbol{g}] = (\boldsymbol{m}(\boldsymbol{x})^T \boldsymbol{s}_p(\boldsymbol{x}))^T \boldsymbol{f}(\boldsymbol{x}) + (\nabla_{\boldsymbol{x}}^T \boldsymbol{m}(\boldsymbol{x}))^T \boldsymbol{f}(\boldsymbol{x})$$
$$+ Tr[\boldsymbol{m}(\boldsymbol{x}) \nabla_x \boldsymbol{f}(\boldsymbol{x})]$$
$$= (\boldsymbol{m}(\boldsymbol{x})^T \boldsymbol{s}_p(\boldsymbol{x}))^T \boldsymbol{f}(\boldsymbol{x}) + \nabla_{\boldsymbol{x}}^T [\boldsymbol{m}(\boldsymbol{x}) \boldsymbol{f}(\boldsymbol{x})] \tag{22}$$

which is exactly the same as the inner part of DSD (Eq.7). So with change of variable formula, we can easily show

$$\mathcal{S}(q_Y, p_Y) = DSD_m(q_X, p_X) \tag{23}$$

$\square$

## B. Proof of proposition 3.1

With the definition of the Riemannian manifold $(\mathcal{M}, \boldsymbol{g})$, for any point $\boldsymbol{a} \in \mathcal{M}$ with local coordinates $\boldsymbol{x} \in \mathbb{R}^D$, and two vectors $\boldsymbol{u}, \boldsymbol{v}$ from its tangent plane $T_a \mathcal{M}$, we can represents $\boldsymbol{u}, \boldsymbol{v}$ using the basis $(\frac{\partial}{\partial x_i})_{\boldsymbol{a}}$ as

$$\boldsymbol{u} = \sum_{i=1}^D u_i (\frac{\partial}{\partial x_i})_{\boldsymbol{a}}, \quad \boldsymbol{v} = \sum_{i=1}^D v_i (\frac{\partial}{\partial x_i})_{\boldsymbol{a}} \tag{24}$$

The inner product defined by the metric $\boldsymbol{g}$ can be expressed as

$$\boldsymbol{g}(\boldsymbol{u}, \boldsymbol{v}) = \sum_{i,j}^D u_i v_j \langle (\frac{\partial}{\partial x_i})_{\boldsymbol{a}}, (\frac{\partial}{\partial x_j})_{\boldsymbol{a}} \rangle_g = \sum_{i,j}^D u_i g_{ij}(\boldsymbol{x}) v_j \tag{25}$$

where $g_{ij}(\boldsymbol{x})$ is the $ij-$th element of matrix $\boldsymbol{G}(\boldsymbol{x})$ and $\langle \cdot, \cdot \rangle_g$ is the inner product defined by Riemannian metric $\boldsymbol{g}$.

We assume the measure $\mathcal{M}(\boldsymbol{x})$ is absolutely continuous w.r.t. Lebesgue measure, then we have the following change of variable formula

$$d\mathcal{M}(\boldsymbol{x}) = \sqrt{|\boldsymbol{G}(\boldsymbol{x})|} d\boldsymbol{x} \tag{26}$$

Then we can represents the densities $\tilde{p}, \tilde{q}$ under Lebessgue measure

$$p(\boldsymbol{x}) = \frac{d\mathbb{P}}{d\mathcal{M}(\boldsymbol{x})} \frac{d\mathcal{M}(\boldsymbol{x})}{d\boldsymbol{x}} = \tilde{p}(\boldsymbol{x}) \sqrt{|\boldsymbol{G}(\boldsymbol{x})|} \tag{27}$$

and $q(\boldsymbol{x})$ is defined accordingly. The score matching loss for $\tilde{p}$ and $\tilde{q}$ is

$$\mathcal{F}_{\mathcal{M}}(\tilde{q}, \tilde{p}) = \frac{1}{2} \int \tilde{q}(\boldsymbol{x}) ||\nabla \log \tilde{p}(\boldsymbol{x}) - \nabla \log \tilde{q}(\boldsymbol{x})||_g^2 d\mathcal{M}(\boldsymbol{x})$$
$$= \frac{1}{2} \int q(\boldsymbol{x}) ||\nabla \log \tilde{p}(\boldsymbol{x}) - \nabla \log \tilde{q}(\boldsymbol{x})||_g^2 d\boldsymbol{x} \tag{28}$$

Now let's define $\nabla \log \tilde{p}(\boldsymbol{x})$. From the basics of Riemannian manifold, for a point $\boldsymbol{a} \in \mathcal{M}$ with local coordinate $\boldsymbol{x}$, and $\mathcal{X}$ is a vector field on $\mathcal{M}$, we have the following definition

$$\langle \sum_{i=1}^{D}(\nabla \log \tilde{p}(\boldsymbol{x}))_i (\frac{\partial}{\partial x_i})_{\boldsymbol{a}}, \sum_{j=1}^{D} \mathcal{X}_j (\frac{\partial}{\partial x_j})\rangle_g = \sum_{i=1}^{D} \mathcal{X}_i \frac{\partial \log \tilde{p}}{\partial x_i} \tag{29}$$

Written in terms of matrix form, assume $\boldsymbol{X} = [\mathcal{X}_1, \ldots, \mathcal{X}_D]^T$, and $g_{ij}(\boldsymbol{x})$ is the element of symmetric positive definite matrix $\boldsymbol{G}(\boldsymbol{x})$, we have

$$(\nabla \log \tilde{p})^T \boldsymbol{G}(\boldsymbol{x})\boldsymbol{X} = (\frac{\partial \log \tilde{p}}{\partial \boldsymbol{x}})^T \boldsymbol{X}$$
$$\Longrightarrow \nabla \log \tilde{p} = \boldsymbol{G}^{-1}(\boldsymbol{x})(\frac{\partial \log \tilde{p}}{\partial \boldsymbol{x}}) \tag{30}$$

Therefore, we have

$$||\nabla \log \tilde{p}(\boldsymbol{x}) - \nabla \log \tilde{q}(\boldsymbol{x})||_g^2$$
$$=\langle \nabla \log \tilde{p}(\boldsymbol{x}) - \nabla \log \tilde{q}(\boldsymbol{x}), \nabla \log \tilde{p}(\boldsymbol{x}) - \nabla \log \tilde{q}(\boldsymbol{x})\rangle_g$$
$$=\langle \boldsymbol{G}^{-1}(\boldsymbol{x}) \underbrace{(\frac{\partial \log \tilde{p}}{\partial \boldsymbol{x}} - \frac{\partial \log \tilde{q}}{\partial \boldsymbol{x}})}_{\tilde{\Delta}(\boldsymbol{x})}, \boldsymbol{G}^{-1}(\boldsymbol{x})(\frac{\partial \log \tilde{p}}{\partial \boldsymbol{x}} - \frac{\partial \log \tilde{q}}{\partial \boldsymbol{x}})\rangle_g$$
$$=\tilde{\Delta}(\boldsymbol{x})^T \boldsymbol{G}^{-1}(\boldsymbol{x})\boldsymbol{G}(\boldsymbol{x})\boldsymbol{G}^{-1}(\boldsymbol{x})\tilde{\Delta}(\boldsymbol{x})$$
$$=\tilde{\Delta}(\boldsymbol{x})^T \boldsymbol{G}^{-1}(\boldsymbol{x})\tilde{\Delta}(\boldsymbol{x}) \tag{31}$$

By change of variable formula, it is also easy to show that

$$\tilde{\Delta}(\boldsymbol{x}) = \underbrace{(\frac{\partial \log p}{\partial \boldsymbol{x}} - \frac{\partial \log q}{\partial \boldsymbol{x}})}_{\Delta(\boldsymbol{x})} \tag{32}$$

Therefore, we have

$$||\nabla \log \tilde{p}(\boldsymbol{x}) - \nabla \log \tilde{q}(\boldsymbol{x})||_g^2 = \Delta^T(\boldsymbol{x})\boldsymbol{G}^{-1}(\boldsymbol{x})\Delta(\boldsymbol{x}) \tag{33}$$

Substitute back to $\mathcal{F}_\mathcal{M}(\tilde{q}, \tilde{p})$ (Eq.28), we can obtain the result. Particularly, compare to *diffusion Fisher divergence* (Eq.5), we can observe that if $\boldsymbol{G}(\boldsymbol{x}) = \boldsymbol{m}(\boldsymbol{x})^{-T}\boldsymbol{m}(\boldsymbol{x})^{-1}$, the $\mathcal{F}_\mathcal{M}(\tilde{q}, \tilde{p})$ is equivalent to *diffusion Fisher divergence*. Indeed, as $\boldsymbol{m}(\boldsymbol{x}) \in \mathbb{R}^{D \times D}$ is an invertible matrix, then $\boldsymbol{m}(\boldsymbol{x})^{-T}\boldsymbol{m}(\boldsymbol{x})^{-1}$ must be symmetric positive definite, which satisfies the requirements for $\boldsymbol{G}(\boldsymbol{x})$.

## C. Proof of proposition 3.2

An ODE flow is defined by the solution of an ODE:

$$d\boldsymbol{x} = \boldsymbol{g}(\boldsymbol{x})dt \tag{34}$$

with $\boldsymbol{g}(\boldsymbol{x})$ the deterministic drift term. Let us consider the forward Euler discretisation of the ODE, which gives

$$\boldsymbol{x}(t+\delta) = \boldsymbol{x}(t) + \delta \boldsymbol{g}(\boldsymbol{x}(t)) := \boldsymbol{T}_\delta(\boldsymbol{x}(t)). \tag{35}$$

With $\delta \approx 0$ we see that $T_\delta$ is an invertible transformation. Now consider $\mathbf{y} = \boldsymbol{x}(t+\delta)$ and $\boldsymbol{x}(t) = \boldsymbol{x}$. This again pushes $p_X(\boldsymbol{x})$ and $q_X(\boldsymbol{x})$ to $p_Y(\mathbf{y})$ and $q_Y(\mathbf{y})$, respectively. Therefore we can reuse results from theorem 3.1 and derive

$$\mathcal{F}(p_Y, q_Y) = \frac{1}{2}\mathbb{E}_{q_X}[||\boldsymbol{m}(\boldsymbol{x})^T(\boldsymbol{s}_{p_X}(\boldsymbol{x}) - \boldsymbol{s}_{q_X}(\boldsymbol{x}))||^2], \tag{36}$$

where $\boldsymbol{m}(\boldsymbol{x}) = (\nabla_{\boldsymbol{x}}\boldsymbol{T}_\delta(\boldsymbol{x}))^{-1}$ Notice that $T_\delta(\boldsymbol{x}) = \boldsymbol{x}$ when $\delta = 0$. This means we can compute the change of score matching at time $t$ as:

$$\frac{\partial}{\partial t}\mathcal{F}(p_Y, q_Y) = \lim_{\delta \to 0^+} \frac{F(p_Y, q_Y) - F(p_X, q_X)}{\delta}$$
$$= \frac{1}{2}\lim_{\delta \to 0^+} \mathbb{E}_{q_X(\boldsymbol{x})}[\Delta(\boldsymbol{x})^\top \delta^{-1}(m(\boldsymbol{x})m(\boldsymbol{x})^\top - \mathbf{I})\Delta(\boldsymbol{x})], \tag{37}$$

with $\Delta(\boldsymbol{x}) = \nabla_{\boldsymbol{x}}\log p_X(\boldsymbol{x}) - \nabla_{\boldsymbol{x}}\log q_X(\boldsymbol{x})$. As $\nabla_{\boldsymbol{x}}T_\delta(\boldsymbol{x}) = \mathbf{I} + \delta\nabla_{\boldsymbol{x}}\boldsymbol{g}(\boldsymbol{x})$, simple calculation shows that

$$\delta^{-1}(m(\boldsymbol{x})m(\boldsymbol{x})^\top - \mathbf{I})$$
$$=\delta^{-1}[[(\mathbf{I} + \delta\nabla_{\boldsymbol{x}}\boldsymbol{g}(\boldsymbol{x}))^\top(\mathbf{I} + \delta\nabla_{\boldsymbol{x}}\boldsymbol{g}(\boldsymbol{x}))]^{-1} - \mathbf{I}]$$
$$=\delta^{-1}[[\mathbf{I} + \delta(\nabla_{\boldsymbol{x}}\boldsymbol{g}(\boldsymbol{x}) + \nabla_{\boldsymbol{x}}\boldsymbol{g}(\boldsymbol{x})^\top) + \mathcal{O}(\delta^2)]^{-1} - \mathbf{I}]$$
$$= -[\nabla_{\boldsymbol{x}}\boldsymbol{g}(\boldsymbol{x}) + \nabla_{\boldsymbol{x}}\boldsymbol{g}(\boldsymbol{x})^\top + \mathcal{O}(\delta)]$$
$$[\mathbf{I} + \delta(\nabla_{\boldsymbol{x}}\boldsymbol{g}(\boldsymbol{x}) + \nabla_{\boldsymbol{x}}\boldsymbol{g}(\boldsymbol{x})^\top) + \mathcal{O}(\delta^2)]^{-1}, \tag{38}$$

which leads to

$$\frac{\partial}{\partial t}\mathcal{F}(p_Y, q_Y)$$
$$= \lim_{\delta \to 0^+} \frac{1}{2}\mathbb{E}_{q_X(\boldsymbol{x})}[\Delta(\boldsymbol{x})^\top \delta^{-1}(m(\boldsymbol{x})m(\boldsymbol{x})^\top - \mathbf{I})\Delta(\boldsymbol{x})]$$
$$= -\frac{1}{2}\mathbb{E}_{q_X(\boldsymbol{x})}[\Delta(\boldsymbol{x})^\top(\nabla_{\boldsymbol{x}}\boldsymbol{g}(\boldsymbol{x}) + \nabla_{\boldsymbol{x}}\boldsymbol{g}(\boldsymbol{x})^\top)\Delta(\boldsymbol{x})] \tag{39}$$

As this quantifies the instantaneous changes, replacing $p_t$ and $q_t$ for $p_Y$, $p_X$, $q_Y$ and $q_X$ gives the instantaneous change of score matching loss.

## D. Additional plots

From the motivating example and theorem 3.1, we know $\boldsymbol{m}(\boldsymbol{x}) = (1 + \frac{(\boldsymbol{x}-\theta)^2}{b})$. Therefore, by simple calculus, the corresponding transformation $y = \boldsymbol{T}(\boldsymbol{x})$ can be defined as

$$\mathbf{y} = \boldsymbol{T}(\boldsymbol{x}) = \frac{1}{b\sqrt{b}}\tan^{-1}(\frac{\boldsymbol{x}-\theta}{\sqrt{b}})$$
$$\boldsymbol{x} = \boldsymbol{T}^{-1}(\mathbf{y}) = \sqrt{b}\tan(b\sqrt{b}\mathbf{y}) + \theta \tag{40}$$

Let's define the transformed densities $p_Y(\mathbf{y})$ and $q_Y(\mathbf{y})$ as

$$p_Y(\mathbf{y}) = p(\boldsymbol{T}^{-1}(\mathbf{y}))|\nabla_{\mathbf{y}}\boldsymbol{T}^{-1}(\mathbf{y})|$$
$$q_Y(\mathbf{y}) = q(\boldsymbol{T}^{-1}(\mathbf{y}))|\nabla_{\mathbf{y}}\boldsymbol{T}^{-1}(\mathbf{y})| \tag{41}$$

Therefore, we can plot the log likelihood for the original densities $p,q$ and transformed densities $p_Y$, $q_Y$ as Figure 2.

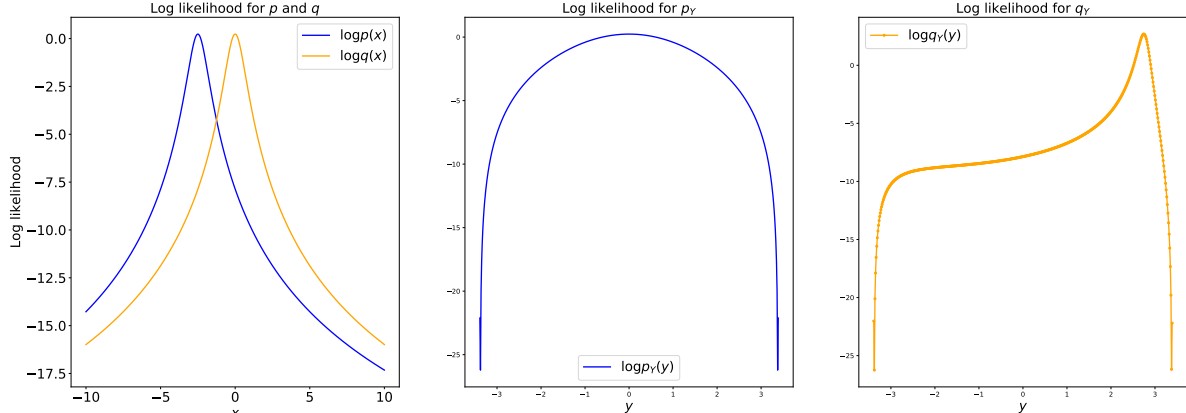

*Figure 2.* **Left**: The log likelihood plot for original densities $q, p$. **Middle**: The log likelihood function for transformed density $p_Y$ **Right**: The log likelihood function for $q_Y$. We choose $\theta = -2.5$ and $b = 0.6$. Notice that the transformed densities $p_Y$, $q_Y$ are periodic as we consider $y \in \mathbb{R}$. This won't happen if we consider $\boldsymbol{y} = \boldsymbol{T}(\boldsymbol{x})$. Because all $\boldsymbol{x}$ value will be squeezed inside the period containing 0, i.e. $\boldsymbol{y}$ will inside $[-3.37, 3.37]$ in this case.

In this case, we set $p(\boldsymbol{x})$ has the mean $-2.5$ with the same scale $0.3$ as $q$, whereas $q$ has mean $0$. For the transformation $\boldsymbol{T}$, we set $\theta = -2.5$ with $b = 0.6$.

# E. Robustness of Gaussian flow

Here, we investigate the robustness of the diffusion matrix w.r.t. the degree-of-freedom (DoF) of studnet-t distribution. We adopt the similar settings as the motivating example (Section 2.2) where $q$ and $p_\theta$ are Student-t distribution with same scale parameter. We vary their DoF together to investigate the changes in DSM loss. Because the manual flow lacks a proper interpretation, so it is difficult to adapted to the change of DoF. Thus, we use the same $\boldsymbol{m}(\boldsymbol{x}) = 1 + \frac{(\boldsymbol{x}-\theta)^2}{0.6}$ for all DoF. On the other hand, Gaussian flow is designed to transform from Student-t to standard Gaussian. So it can be easily adapted to the change of DoF by using the corresponding $F_\theta$.

Figure 3 plots the DSM losses with both manual flow and Gaussian flow. From the top panel, we can clearly observe that manual flow only works with DoF $= 5$. For other DoF, the corresponding DSM fails to recover the ground truth $\theta$. On the other hand, the bottom panel shows that Gaussian flow is robust to the change of DoF, and consistently gives the correct ground truth $\theta$ with wide fast convergence region.

We emphasize again that for both flows, the resulting $DSM_m(q, p_\theta)$ is not equivalent to $\mathcal{F}_m(q, p_\theta)$ (Eq. 5) as $C_{q,m}$ in Eq. 6 is dependent on $\theta$. However, unlike the manually designed flow by Barp et al. (2019), the Gaussian flow returns surrogate losses that have only one global optimum at the desired solution in all cases considered. Future work will evaluate the Gaussian flow construction of DSM objectives beyond student-t case for further understandings.

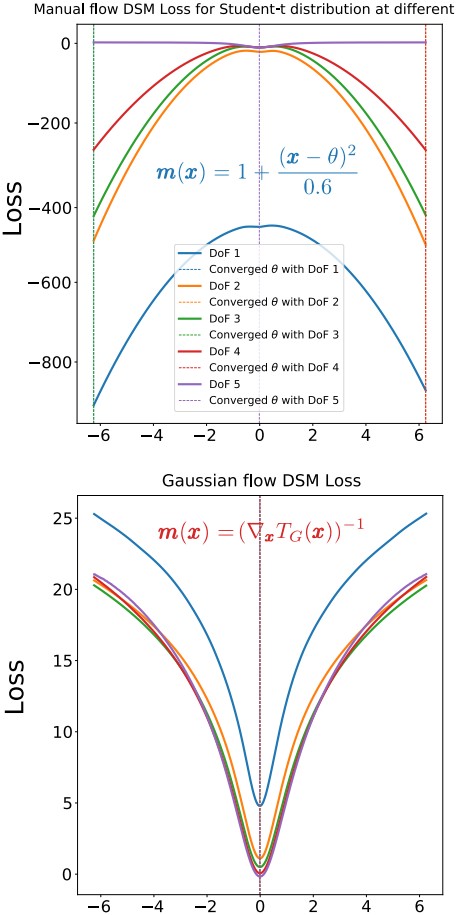

*Figure 3.* The $DSM_m(q, p_\theta)$ losses using $p_\theta$, $q$ with different degree-of-freedom (DoF). They are plotted when $m(\boldsymbol{x})$ is constructed using the manual flow (top) or Gaussian flow (bottom).