# OpenReview forum: "Interpreting diffusion score matching using normalizing flow"
_ICML.cc/2021/Workshop/INNF — INNF+ 2021 spotlighttalk_

### Official Review · Reviewer_yuH9 · 2021-06-09

**Rating:** Accept
**Confidence:** 2

**Summary:**

In this paper, the authors show the connections between diffusion score matching, diffusion Stein discrepancy and their original formats using normalizing flows.

**Justification For Rating:**

This is a theoretical paper. I checked some of the proof and did not find any issues. One minor suggestion is the authors can slightly discuss how to apply these relationships to practical problems.

---

### Official Review · Reviewer_7c4B · 2021-06-11

**Rating:** Accept
**Confidence:** 5

**Summary:**

Diffusion score matching (DSM) and diffusion Stein discrepancy (DSD) solve some shortcomings of SM and SD. A proper choice of diffusion matrix is critical, and yet interpretation of such matrix is missing. This theoretical paper proves that given a distribution defined under a normalizing flow framework, the two are equivalent since the diffusion matrix is the inverse Jacobian matrix. Additional interpretation is provided under Riemannian manifold.


**Justification For Rating:**

This paper gives crucial insights into the interpretation of the diffusion matrix, necessary to select a proper matrix for an estimator. It's an interesting theoretical finding to conclude the diffusion matrix matches the inverse of the flow Jacobian. Further work could build on these findings to design DSM/DSD diffusion matrices. On the flipside, how do normalizing flows benefit from this finding? What are the theoretical conclusions about flows knowing the inverse Jacobian acts like a diffusion matrix? It would be interesting to leverage findings on designing diffusion matrices to have an optimal inverse Jacobian in normalizing flows.

---

### Decision · Program_Chairs · 2021-06-14

Accept (spotlight talk)